# Observation of a topological nodal surface and its surface-state arcs in an artificial acoustic crystal

Yihao Yang [1,2,5], Jian-ping Xia[3,5], Hong-xiang Sun [3*], Yong Ge[3], Ding Jia[3], Shou-qi Yuan[3], Shengyuan A. Yang [4], Yidong Chong [1,2*] & Baile Zhang [1,2*]

Three-dimensional (3D) gapless topological phases can be classified by the dimensionality of the band degeneracies, including zero-dimensional (0D) nodal points, one-dimensional (1D) nodal lines, and two-dimensional (2D) nodal surfaces. Both nodal points and nodal lines have been realized recently in photonics and acoustics. However, a nodal surface has never been observed in any classical-wave system. Here, we report on the experimental observation of a twofold symmetry-enforced nodal surface in a 3D chiral acoustic crystal. In particular, the demonstrated nodal surface carries a topological charge of 2, constituting the first realization of a higher-dimensional topologically-charged band degeneracy. Using direct acoustic field measurements, we observe the projected nodal surface and its Fermi-arc-like surface states and demonstrate topologically-induced robustness of the surface states against disorders. This discovery of a higher-dimensional topologically-charged band degeneracy paves the way toward further explorations of the physics and applications of new topological semimetal phases.

[1] Division of Physics and Applied Physics, School of Physical and Mathematical Sciences, Nanyang Technological University, 21 Nanyang Link, Singapore 637371, Singapore. [2] Centre for Disruptive Photonic Technologies, The Photonics Institute, Nanyang Technological University, 50 Nanyang Avenue, Singapore 639798, Singapore. [3] Research Center of Fluid Machinery Engineering and Technology, Faculty of Science, Jiangsu University, Zhenjiang 212013, China. [4] Research Laboratory for Quantum Materials, Singapore University of Technology and Design, Singapore 487372, Singapore. [5] These authors contributed equally: Yihao Yang, Jian-ping Xia. *email: jsdxshx@ujs.edu.cn; yidong@ntu.edu.sg; blzhang@ntu.edu.sg

Specially designed structures including metamaterials and artificial crystals in classical-wave systems can possess topological properties, which offer fundamentally new degrees of freedom to manipulate the classical waves in photonics, acoustics, and mechanics[1–5]. The most representative examples are the photonic/acoustic analogs of topological insulators (generally referred to as "photonic/acoustic topological insulators") that behave as "insulating" materials in bulk but support chiral or helical states on their edges[2,6–9]. An important property of such edge states is that they are robust against backscattering caused by disorders and defects, which is guaranteed by the nontrivial band topology. These photonic/acoustic topological insulators have found a lot of applications, including backscattering-immune waveguides[9,10], robust delay lines[11], and topological waveguide splitters[12,13]. The discovery of the photonic/acoustic topological insulators has soon stimulated broad research interest, giving birth to the emerging fields of topological photonics/acoustics[1,2,4,5].

Unlike topological insulators, which have band gaps, topological semimetals are gapless, and their topological properties are characterized by band degeneracies[14,15]. Such gapless topological phases have also found their classical analogs in photonics and acoustics[4,5]. Weyl photonic/acoustic crystals[16–22], for example, host twofold zero-dimensional (0D) point degeneracies known as Weyl points which are monopoles of quantized Berry flux in three-dimensional (3D) momentum space and carry topological charge ±1; when projected to a surface, two oppositely charged Weyl points are connected in momentum space by open surface-state arcs. As it turns out, Weyl points are only one member of a broader family of topologically nontrivial band degeneracies. Researchers have demonstrated photonic/acoustic crystals with topological nodal lines—band degeneracies lying along one-dimensional (1D) curves in momentum space—which are associated with Berry phase winding of π and drumhead surface states[23–26]. Recent theoretical studies have also revealed the possibility of higher-dimensional topological band degeneracies, such as twofold two-dimensional (2D) nodal surfaces[27–34]. Along the normal of nodal surfaces, the Bloch states possess a pseudospin degree of freedom associated with the band index and have an effective description in terms of massless 1D Dirac particles[30]. Interestingly, in the acoustic/photonic systems[29,33], there are two linear Weyl points pared with the topological nodal surface and the total topological charges of all bands are zero, being consistent with the Nielsen–Ninomiya theorem[35]. In comparison to the condensed matter systems where the nodal surfaces usually are fragile to spin–orbit interaction effects[30] or coexist with the other trivial bands, the classical-wave systems are advantageous in realizing the 2D nodal surfaces, as the structure of each unit cell can be precisely tailored and the resulting band-structure is easily probed at various frequencies, and also because classical waves experience no intrinsic spin–orbit coupling. Despite the great interest in topological nodal surfaces, there has been no experimental observation in any classical-wave system to date[29,33].

This work reports on the experimental demonstration of a twofold topological nodal surface in a chiral 3D phononic crystal. The nodal surface is stabilized by a non-symmorphic lattice symmetry and time-reversal symmetry and takes the form of a nodal plane spanning the Brillouin zone (i.e., a torus in momentum space). As described below, the nodal surface carries a topological charge of 2, associated with inversion symmetry breaking in the underlying lattice. To the best of our knowledge, this is the first realization of a nodal surface in classical-wave systems, and also the first realization of a higher-dimensional topologically charged band degeneracy beyond the previously discovered topologically charged 0D points including the Weyl points and their variants in all physical systems[15,29]. The nodal surface exhibits a topological bulk-edge correspondence in the form of a pair of open arcs that emanate from the projected nodal surface, connecting to two Weyl cones elsewhere in the projected Brillouin zone. Using acoustic measurements, we directly image the projected nodal surface and its Fermi-arc-like surface states and demonstrate the robustness of surface states against disorder.

## Results

**Design of a 3D acoustic crystal with the topological nodal surface.** The 3D artificial acoustic crystal, fabricated by 3D printing, is shown in Fig. 1a, b. The 3D hexagonal unit cell, which has three length parameters $h$, $r$, and $t$, is shown in Fig. 1c. Here, the background is solid resin, with three types of hollow air tubes indicated by three different colors. The blue tubes run along the $z$-direction; the red tubes connect next-nearest-neighbor sites on the $z = h/2 + n \times h$ planes (where $n$ is an integer), and the green tubes connect next-nearest-neighbor sites on the $z = n \times h$ planes. Note that the green and red tubes do intersect. Acoustic (sound) waves propagate in the network of air tubes without penetrating the solid resin background. The crystal has the non-symmorphic space group symmetry P6$_3$ (No. 173), which lacks inversion, mirror, or other roto-inversion symmetries; it is a so-called topological chiral crystal[32,36–39]. The resulting acoustic crystal has twofold screw rotations along the $z$-direction $S_{2z}$: $(x, y, z) \rightarrow (-x, -y, z + 0.5\,h)$, as well as bosonic time-reversal symmetry ($T^2 = 1$). The combination $TS_{2z}$ satisfies $(TS_{2z})^2 = e^{-ik_z h}$, where $k_z$ is the wavevector in the $z$-direction. Along the $k_z = \pi/h$ plane, $(TS_{2z})^2 = -1$. Therefore, twofold Kramers degeneracy applies at every point on this plane, forming a topological nodal surface[28–30].

The 3D Brillouin zone and the corresponding numerically calculated acoustic band structures are shown in Fig. 1d, e. There is a nodal surface along the $k_z = \pi/h$ plane, shown in the right inset of Fig. 1f. Away from the topological nodal surface, the dispersion is highly linear, as shown by the R–Σ and K–H high-symmetry lines in Fig. 1e. The nodal surface is stabilized by the twofold screw rotation and time-reversal symmetries, and its nonzero topological charge is specifically due to the lack of inversion symmetry[29,32,33].

To find the topological charge, we compute the Chern numbers of the acoustic Bloch wavefunctions on a surface enclosing the topological nodal surface using the Wilson loop method[32]. This first-principles calculation reveals that the topological charge of the nodal surface is +2. Accompanying the topological nodal surface, there exist a pair of Weyl points at K/K′ (marked as red dots in Fig. 1d, e, and shown in the left inset of Fig. 1f), which both have topological charge −1, in agreement with the Nielsen–Ninomiya theorem which states that the topological charges must sum to zero[35]. For details of these calculations, refer to Supplementary Note 1. Over the frequency range 6.15–7.9 kHz, indicated by the horizontal red lines in Fig. 1e, Fermi-arc-like surface states can be observed, as described below. Besides, we should note that in comparison to Xiao et al.'s design[29], our design has two advantages that facilitate experiment. Firstly, our design has a very large frequency range where the topological surface states exist (relative bandwidth ~20%), while the design in ref. [29] is small (relative bandwidth <2%). Secondly, there are z-oriented tubes running through the sample. The acoustic probes then can be inserted into the sample through these tubes, allowing the direct characterization of bandstructure.

**Experimental observation of the topological nodal surface and its surface-state arcs.** We perform three experiments to characterize the topological properties of the topological nodal surface (see experimental details in Methods). First, we measure the projected bulk bandstructure by probing the sound pressure along a plane passing through the middle of the sample, as shown

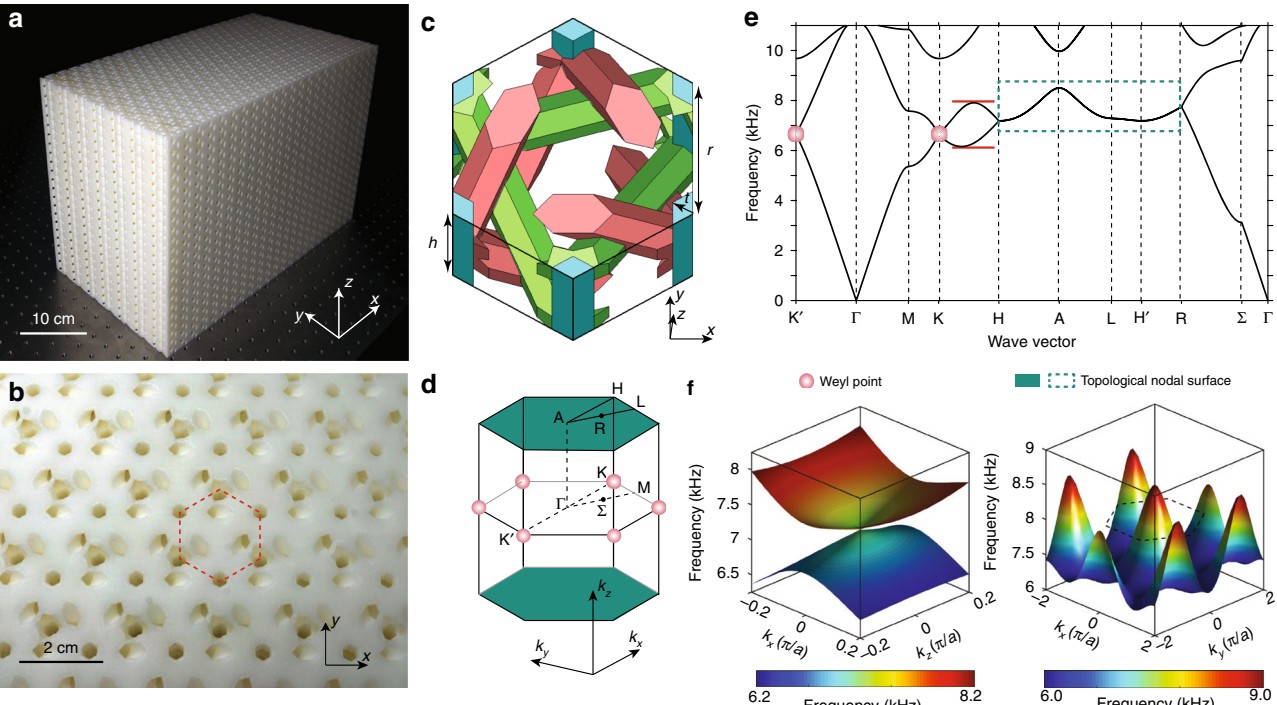

**Fig. 1** A 3D artificial acoustic crystal with a topological nodal surface and two Weyl points. **a**, **b** Perspective and top views of the fabricated 3D acoustic crystal sample, consisting of 23 × 13 × 20 unit cells. The dashed hexagon in (**b**) denotes a unit cell. **c** Schematic of a unit cell. The background is solid resin, and blue, red, and green regions represent hexagonal air tubes which have identical cross-sectional profiles. Here, $h = 15$ mm, $r = 11.55$ mm, and $t = 2$ mm. **d**, **e** 3D Brillouin zone and bandstructure of the acoustic crystal. The bands enclosed by the green dashes form the topological nodal surface, and red dots indicate the two additional Weyl points. The horizontal red lines indicate the topologically nontrivial frequency range where Fermi-arc-like surface states can exist. (**f**) 2D band structures around the Weyl points and the topological nodal surface. Each Weyl point has a topological charge of −1, and the topological nodal surface has a topological charge of +2, revealed via our numerical calculations (see Supplementary Note 1)

in Fig. 2a. Here, both the probe and the source are inserted into the sample via the vertical air tubes indicated in blue in Fig. 1c. The point source is located 4 cm away from the measurement plane, and therefore the sound waves measured along the plane carry both in-plane and out-of-plane momenta. We apply spatial Fourier transforms to the measured complex acoustic field distributions, and hence obtain the projected bulk bandstructure shown in Fig. 2c. For comparison, the numerically calculated projected band diagram is shown in Fig. 2d. As can be seen, there is a good quantitative agreement between experimental and numerical results, and the projections of the nodal surface and Weyl points are clearly visible in both sets of results.

Next, we probe for surface waves at a boundary between the acoustic crystal and acrylic board. The experimental configuration is shown in Fig. 3a. In the results shown in Fig. 3b, and the corresponding numerical results of Fig. 3c, we observe a family of surface states in the 6.0 to 8.0 kHz frequency range (see Fig. 1e). Note that the slight difference between the measured (Fig. 3b) and simulated (Fig. 3c) results is due to the experimental error caused by a limited resolution in momentum space. The numerical isofrequency plots in Fig. 3d, f indicate that there are two surface states that emanate from the projected nodal surface, each connecting to one of the two projected Weyl cones. This is consistent with the aforementioned fact that the nodal surface possesses a nontrivial topological charge of +2. Note that due to the time-reversal symmetry preserved in our system, we have employed the inversion symmetry operator to make the measured dispersion more symmetric. The experimental results, shown in Fig. 3e, g, are in good agreement. Besides, due to the limited accuracy of the fabrication and measurement, there are slight differences between the simulated and measured results. As the

frequency decreases and the corresponding wavelength increases, the acoustic dispersion becomes more insensitive to the details. Therefore, the agreement between simulation and measured data look better at lower frequencies.

Finally, we probe the robustness of the Fermi-arc-like surface states. As indicated in Fig. 4a, iron rods of radius 1.5 mm are inserted into the vertical air tubes in a region of the lattice adjacent to the boundary with the acrylic board. The iron rods block the flow of sound within the affected tubes, which are the principal route over which energy can pass between different layers of the structure (see Fig. 1c); hence, they should have a dramatic effect on the acoustic modes. Surface waves are then launched from one side of the disorder region, and the sound pressure is measured along the $y = 0$ mm plane. The results in Fig. 4b clearly show the surface waves passing through the disorder region. A spatial Fourier transform is applied to these results, producing the momentum space plot shown in Fig. 4c. This reveals that the acoustic waves are predominantly concentrated in the forward direction, with negligible back-reflection from the disorder region. It can be well explained as follows: in our experiment, the excitation is optimized and almost only the open-arc surface states with $kz$ located in $[0.1\pi/pz, 0.75\pi/pz]$ are excited at 6.45 kHz. In this range, one can view each $kx, ky$ plane for a fixed $kz$ (preserved as a result of translational symmetry along the $z$ axis) as a 2D Chern insulator[15] (see Fig. 3d, e). Therefore, the back-reflection is strongly suppressed.

## Discussion

We have experimentally observed a twofold topological nodal surface stabilized by the symmetries of a 3D non-symmorphic

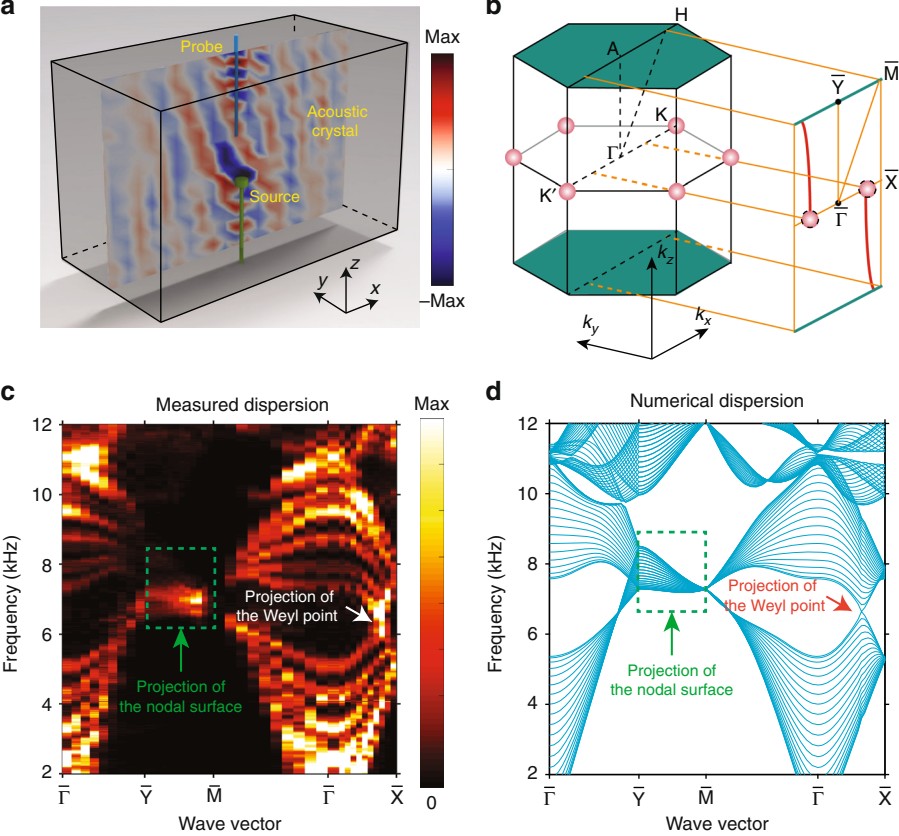

**Fig. 2** Observation of a topological nodal surface and Weyl points. **a** Schematic of the experimental setup. The sound pressure field distribution is measured along an $x$–$z$ plane passing through the middle of the sample, with the source located 4 cm away from the plane in the $y$-direction. **b** Projection of the Brillouin zone onto the 2D measurement plane. The green plane (line) and red dots indicate the projected nodal surface and Weyl points, respectively. The red lines denote the surface-state arcs that connect the topological nodal surface to each Weyl point. **c**, **d** Measured and simulated projected band spectra. The color bar in (**c**) indicates the energy intensity

acoustic crystal. Our results clearly demonstrate that the nodal surface carries a topological charge of 2, and therefore constitutes the first realization of topologically charged band degeneracy beyond the previously realized Weyl points and their variants. We observe topological bulk-edge correspondence in the form of paired Fermi-arc-like surface states, which connect the topological nodal surface to two different Weyl points. The present acoustic crystal provides an excellent platform for studying the topological nodal surface phenomenon since the surface-state arcs exist over a relatively broad (~20%) frequency range. In future work, similar acoustic crystals may be used to explore other types of band degeneracies. It would be interesting to perform a detailed study of the topological nodal surface states, due to the possibility of realizing exotic one-way chiral zero modes that cannot be achieved with standard Weyl points[40,41].

## Methods

**Numerical methods**. All simulations are performed in the acoustic module of commercial finite element method software COMSOL Multiphysics. Due to the large impedance difference between air and the photosensitive resin the surfaces of the resin region are simulated as hard boundaries. Besides, due to the limited sound pressure in our experiment, the nonlinear effects are negligible, as manifested in previous works[9,19,21]. The air density is 1.18 kg m$^{-3}$ and the speed of sound is 343 m s$^{-1}$. To calculate the bulk dispersion of the unit cell, periodic boundary conditions are applied in all directions. To calculate the surface wave modes, periodic boundary conditions are applied in the $x$ and $z$ directions, with a hard boundary in the $y$-direction. The simulations use only 10 unit cells in the $y$-direction, which is sufficient since the surface waves are extremely well confined.

**Sample fabrication**. The sample is manufactured via an additive manufacturing technique (stereolithography). The material is photosensitive resin with modulus 2880 MPa and density 1.10 g cm$^{-3}$.

**Experimental setup**. In the surface-state measurement, a square acrylic plate (length of 500 mm), whose surfaces act as hard boundaries, covers the right side of the sample. A balanced armature speaker (radius of about 1 mm) driven by a power amplifier, working as a broadband sound source, is placed at the center of the interface between the acrylic plate and the sample. For the bulk state measurement, the broadband sound signal is launched from a narrow tube (radius of about 1.5 mm and length of about 200 mm), which penetrates into the sample from sample's bottom. The distance between the tube and the right surface is about 100 mm.

To measure acoustic pressure fields, two microphones (radius of about 3.2 mm, Brüel&Kjær Type 4961) are separately placed in a sealed sleeve with a tube (radius of 1 mm and length of 350 mm) that penetrates deep into the sample During each measurement, microphone 1 is scanned point-by-point through one of the vertical air holes of the structure, to detect the input acoustic signal. Microphone 2 is fixed towards the sound source to detect the reference acoustic signal. The scanning steps in the $x$ and $z$ directions are 20 mm and 15 mm, respectively. The distances between the measured plane and the right surface of the sample are about 5 mm and 140 mm for the surface and bulk states, respectively. Both the amplitude and phase of the acoustic pressure field are recorded by Brüel&Kjær 3160-A-022 module. The bulk and surface-state dispersions are obtained by applying Fourier transform to the measured complex acoustic pressure fields.

## Data availability

The data that support the plots within this paper and other findings of this study are available from the corresponding author upon request.

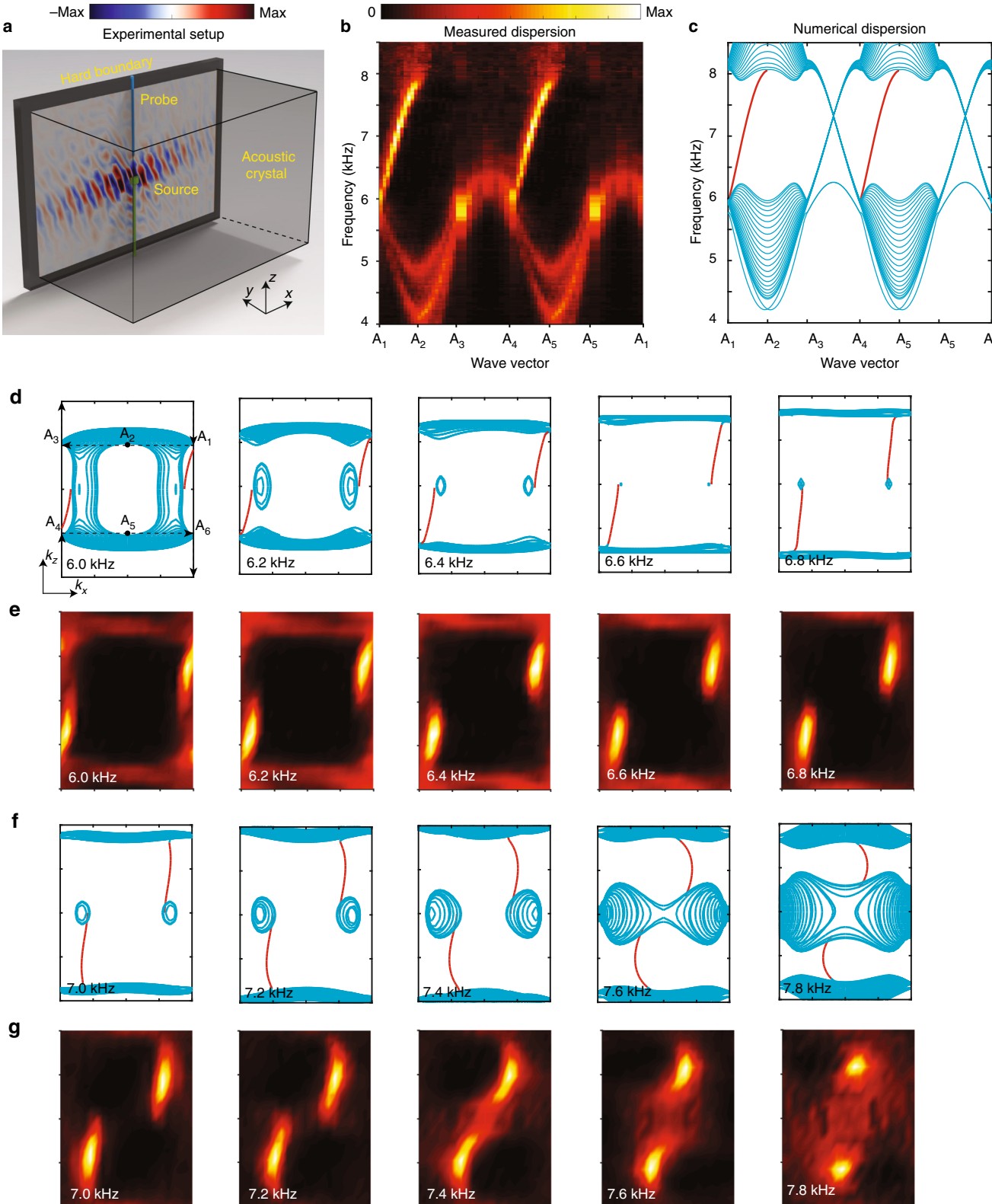

**Fig. 3** Observation of surface-state arcs emanating from the topological nodal surface. **a** Experimental setup. The field distributions are measured along a hard boundary between the acoustic crystal and acrylic board. **b**, **c** Measured and simulated surface dispersion along selected directions in the 2D plane, indicated by the points marked $A_1$–$A_6$ in (**d**). Here, $A_1$ to $A_6$ represent the momentum points $(\pi/a, 0.5\pi/h)$, $(0, 0.5\pi/h)$, $(-\pi/a, 0.5\pi/h)$, $(-\pi/a, -0.5\pi/h)$, $(0, -0.5\pi/h)$, and $(\pi/a, -0.5\pi/h)$, respectively. **d**, **f** Simulated isofrequency surfaces in the 2D momentum space, at specific frequencies ranging from 6.0–7.8 kHz. **e**, **g** Measured in-plane energy densities for the same set of frequencies. For every subplot in (**d**–**g**), the horizontal axis ($k_x$) runs over $[-\pi/a, \pi/a]$, and the vertical axis ($k_z$) runs over $[-\pi/h, \pi/h]$

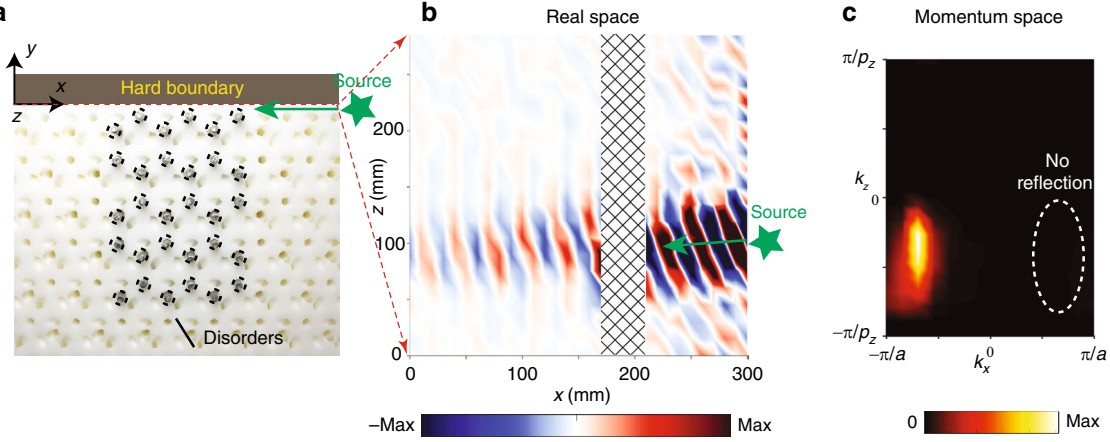

**Fig. 4** Reflection-free propagation of topological Fermi-arc-like surface states. **a** Photograph of the sample with rods inserted into the vertical air holes to serve as a disorder region. The source is located to the right of the photographed region. **b** Measured sound pressure along the $y = 0$ mm plane at 6.45 kHz. The shaded area denotes the disorder region. **c** The energy density of the measured surface waves plotted in momentum space. The dashed ellipse represents where a reflected beam would be located

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

## Acknowledgements

This work was sponsored by the Singapore Ministry of Education under Grants No. MOE2018-T2–1–022 (S), MOE2015-T2–2–008, MOE2017-T2-2-108, MOE2016-T3–1–006, and Tier 1 RG174/16 (S). H.S acknowledges the support of the National Natural Science Foundation of China (11774137). S.Y. acknowledges the support of the National Natural Science Foundation of China (51779107).

## Author contributions

All authors contributed extensively to this work. Y.Y. created the design and performed the simulations. Y.Y. and H.S. fabricated the sample and designed the experiments. H.S., J.X., Y.G., and D.J. performed measurements. Y.Y., S.Y.Y., S.Q.Y., Y.C., and B.Z. analyzed data and wrote the paper. Y.C. and B.Z. supervised the project.

## Competing interests

The authors declare no competing interests.
