## [Peer Review File · Nature Communications]

Reviewers' Comments:

Reviewer #1:

Remarks to the Author:

This work reported the first experiments on a 3D surface degeneracy of a non-zero Chern number in any system. It is quite interesting and convincing.

I recommend its publication with the following minor comments:

- 1) The terminology of "Fermi" in the whole paper is incorrect. It is bosonic and classical. "Surface arc" could work better.
- 2) In Fig. 4c, there should be $-k$ components as the reflection. Not sure the constant k_z argument is clearly explained here.
- 3) Overall, a picture of the measurement setup will be helpful for the readers.

Reviewer #2:

None

Reviewer #3:

Remarks to the Author:

In the manuscript "Observation of a topological nodal surface and its Fermi arcs in an artificial acoustic crystal", the authors present experimental evidences for a topologically charged 2-fold band degeneracy over a 2D surface in the Brillouin zone of a 3D acoustic metamaterial.

The results of the manuscript nicely complete the observation of topological degeneracies in 3D acoustic crystals. These systems are ideal for such studies as they lack of intrinsic spin-orbit coupling which is often detrimental to these extensive degeneracies. However, a similar situation (as stated by the authors) is found in photonic crystals. Ref.[33] in the manuscript (which should be updated as the preprint has now been published) reports an observation of topologically charged nodal surface in a photonic crystal in a similar model. The observations here reported are often portrayed as a primer. I believe these claims should be reformulated to acknowledge the work of Ref.[33]. At the same time, I do not believe that the simultaneous observation of the phenomena in the photonic realm weakens the findings of the manuscript. On the contrary, the observation of the same phenomena in different platforms and different frequencies strengthens its relevance and its potential for future applications.

I have few remarks about the Introduction:

- "Weyl points are connected in momentum space by an unclosed Fermi arcs of topological surface states". It is ambiguous as the Fermi arcs are themselves the topological surface states. I would simply state "Weyl points are connected in momentum space by open Fermi arcs". Maybe, it is also worth to explain once that the Fermi arcs in the context of classical waves (bosonic and not fermionic) are not related to a Fermi surface but just to an equienergy line.

- The advantage of classical platform over real materials is argued to be in the absence of interactions. I do not agree as nonlinearities are the counterpart of interactions for these systems. The authors could elaborate further on why nonlinearities can be neglected in their sample. Alternatively, the concrete advantage of these platforms could be highlighted: the absence of intrinsic spin-orbit coupling.

- The simultaneous presence of Weyl points is briefly mentioned in the introduction. Their necessity is justified only later via the Nielsen-Ninomiya theorem. Such explanation should be provided already in the introduction.

The sample's design is based on the tight-binding model proposed in Ref.[29]. There, it is argued

that a strong direct hopping among different layers can cause a transition from a nodal surface to nodal points. In the manuscript only the radii of the channels mediating the couplings are stated. The small size of the direct channels suggests that the authors are in the right regime of the parameter space. Would it be possible to extract the hopping strength mediated by this channel by mapping the low energy description of the acoustic crystal around the nodal surface to an effective tight binding model?

I assume that in Fig.1f the color of the bands corresponds to the local strength of the Berry curvature field. There should be a colormap to understand the numerical values of this figure. I would be careful in writing a Chern number for a gapless band structure. In the absence of a gap and in 3D the Chern number of a band is not well defined, although it is clear what the authors mean. I think that the colormap with the Berry curvature field value is sufficient for the authors' claim on the topological charge of the Weyl point and the nodal surface.

"we have verified that the depicted surface is indeed twofold degenerate": I find this sentence not useful in the absence of further clarifications on how the degeneracy has been assessed.

Concerning the bulk characterization of the sample: Why did the authors choose to measure the pressure field only along one plane? A full characterization of the sample would have been more convincing. The data can then still be projected on 2d momentum planes or plotted at selected momentum points. In particular, it would have been possible to confirm experimentally the presence of a nodal surface in the entire 2d BZ at $k_z = \pi$. I find the lack of a complete bulk characterization of the crystal the weakest point of the manuscript. I like how for these plots the results of simulations are presented but not overlaid to the measured data.

On the other hand, I find the characterization of the surface states very convincing and exhaustive. I particularly appreciate the experimental study of surface disorder that simultaneously proves the robustness of the surface states and their chirality. Concerning this part of the manuscript, I have only few remarks:

-The exact location of points A1,A2,A3,A4,A5 and A6 in momentum space should be stated. The name suggests that they are at $k_z = \pi$ and $k_z = -\pi$. Their location in k_x is not clear. Have the data been symmetrized?

-Fig.3b: Is there a reason why the measured band seems to extend below 4kHz, while the simulated one ends above it? Is this due to the boundary termination of the sample?

-I think the termination of the sample should be clarified. The sample is closed with a plate and neither zigzag nor bearded termination are realized. Is this termination used also COMSOL?

-Fig.3e&g: As opposed to the bulk measurements, here I would find more helpful if the simulated frequency contour of the projected Weyl points and nodal surface were overlaid to the measured data.

-Why does the agreement between simulation and measured data seem to improve at lower frequencies?

-The frequency step of the measurement is 200Hz. Many figures look similar. Resolving 200Hz would amount to a quality factor of 30/40. What is the estimated quality factor of the sample? Is it reasonable to expect any observable difference over 200Hz?

Finally, there is a typo in the Experiments section of Methods. "The sample is manufactured via fabricated with additive manufacturing technique" should be "The sample is fabricated via additive manufacturing technique".

The manuscript presents neat experimental data that substantiate the observation of a topologically charged nodal surface in acoustic metamaterial. Although a nearly simultaneous observation has been reported in photonic crystals, the results here reported are a significant contribution to the field of topological acoustics. I believe that, provided the issues raised in this report will be addressed, the manuscript meets the criteria of Nature Communications.

Response Letter to Reviewers

We are grateful for the constructive comments on this manuscript (NCOMMS-19-17253) from all the reviewers.

In the text below, each of the comments from each reviewer is quoted in italics and is followed by the corresponding detailed response. We also revised the manuscript and the Supplementary Information accordingly, and these updates are highlighted in blue in those files. In the text below the references to these updates are highlighted in a similar way.

GENERAL COMMENTS FROM 1st REVIEWER:

This work reported the first experiments on a 3D surface degeneracy of a non-zero Chern number in any system. It is quite interesting and convincing.

I recommend its publication with the following minor comments:

Response from Authors:

We thank the reviewer for recommending the publication and considering that “*this work reported the first experiments on a 3D surface degeneracy of a non-zero Chern number in any system. It is quite interesting and convincing.*” In the following, we fully address the specific comments point-by-point.

SPECIFIC COMMENTS FROM 1st REVIEWER:

1st Reviewer -- Comment 1:

1) The terminology of "Fermi" in the whole paper is incorrect. It is bosonic and classical. "Surface arc" could work better.

Response from Authors:

Following the reviewer’s suggestion, we have replaced “Fermi arc” with “surface-state arc” and “Fermi arc surface state” with “Fermi-arc-like surface state” in the revised manuscript.

1st Reviewer -- Comment 2:

2) In Fig. 4c, there should be -k components as the reflection. Not sure the constant k_z argument is clearly explained here.

Response from Authors:

The reviewer’s question regarding Fig. 4c can be divided into two parts: first, why constant k_z is preserved, and second, why there is no reflection. For the first part, the reason why constant k_z is preserved is because the setup in Fig. 4 has translational symmetry along the z axis (the disorders do not break such symmetry). Therefore k_z is preserved as a result of the translational symmetry. For the second part, the reason why the reflection is unnoticeable is because, in our experiment, the excitation is optimized and almost only the open-arc surface states with k_z located in $[0.1\pi/pz, 0.75\pi/pz]$ are excited at 6.45 kHz. In this range, one can view each k_x, k_y plane at a fixed k_z as a 2D Chern insulator (see Fig. 3(d)-(e)). Therefore, the reflection is strongly suppressed.

We have added a discussion on page 10, starting from line 179, which reads as

“It can be well explained as follows: in our experiment, the excitation is optimized and almost only the open-arc surface states with k_z located in $[0.1\pi/pz, 0.75\pi/pz]$ are excited at 6.45 kHz. In

this range, one can view each k_x , k_y plane for a fixed k_z (preserved as a result of translational symmetry along the z axis) as a 2D Chern insulator¹⁵ (see Fig. 3(d)-(e)). Therefore, the back-reflection is strongly suppressed.”

1st Reviewer -- Comment 3:

3) Overall, a picture of the measurement setup will be helpful for the readers.

Response from Authors:

Following the reviewer’s suggestion, we have provided a picture of the measurement setup, as shown in Fig. R1.

Figure R1. Photo of the experimental setup for the surface state measurement. A square acrylic plate covers the measured surface of the sample. A balanced armature speaker as a source is placed at the center of the interface between the acrylic plate and the sample. Two microphones (Microphone 1 and 2) are separately placed in a sealed sleeve with a tube that penetrates deeply into the sample. During each measurement, microphone 1 is scanned point-by-point through one of the vertical air holes of the structure, to detect the input acoustic signal. Microphone 2 is fixed with respect to the sound source in order to detect the reference acoustic signal. The bulk state measurement setup is similar, and the measured plane is located in the middle of the sample.

We have added Figure R1 in the Supplementary Information as Fig. S2.

GENERAL COMMENTS FROM 2nd REVIEWER:

In the manuscript "Observation of a topological nodal surface and its Fermi arcs in an artificial acoustic crystal", the authors present experimental evidences for a topologically charged 2-fold band degeneracy over a 2D surface in the Brillouin zone of a 3D acoustic metamaterial.

The results of the manuscript nicely complete the observation of topological degeneracies in 3D acoustic crystals. These systems are ideal for such studies as they lack of intrinsic spin-orbit coupling which is often detrimental to these extensive degeneracies. However, a similar situation (as stated by the authors) is found in photonic crystals. Ref.[33] in the manuscript (which should

be updated as the preprint has now been published) reports an observation of topologically charged nodal surface in a photonic crystal in a similar model. The observations here reported are often portrayed as a primer. I believe these claims should be reformulated to acknowledge the work of Ref.[33]. At the same time, I do not believe that the simultaneous observation of the phenomena in the photonic realm weakens the findings of the manuscript. On the contrary, the observation of the same phenomena in different platforms and different frequencies strengthens its relevance and its potential for future applications.

...
...

The manuscript presents neat experimental data that substantiate the observation of a topologically charged nodal surface in acoustic metamaterial. Although a nearly simultaneous observation has been reported in photonic crystals, the results here reported are a significant contribution to the field of topological acoustics. I believe that, provided the issues raised in this report will be addressed, the manuscript meets the criteria of Nature Communications.

Response from Authors:

We thank the reviewer for the positive comments and the reminder that Ref. [33] has been published. We have updated Ref. [33] accordingly in the revised manuscript.

We would like to bring to the reviewer's attention that Ref. [33] is a theoretical paper, while our work is experimental. We thank the reviewer's comments that "*I do not believe that the simultaneous observation of the phenomena in the photonic realm weakens the findings of the manuscript...*" However, we would like to make a correction that the phenomena in the photonic realm, as proposed in Ref. [33], have *not* been observed in reality. Therefore, it is our work that firstly reported the experimental observation of such phenomena.

Despite the fundamental difference in terms of experimental observation, our work is indeed partially inspired by Ref. [33] which is one of the earliest feasible proposals for classical-wave constructions [such as Xiao et al.'s proposal for an acoustic construction originally proposed in 2017 (arXiv:1709.02363 (2017); cited as Ref. 29 in our original manuscript)].

In fact, the work of Ref. [33] has been well acknowledged in our original manuscript. The corresponding sentence, on page 3, starting from line 55, reads as "Recent theoretical studies have also revealed the possibility of higher-dimensional topological band degeneracies, such as twofold two-dimensional (2D) nodal surfaces²⁷⁻³⁴."

In the following, we fully address the specific comments point-by-point.

SPECIFIC COMMENTS FROM 2nd REVIEWER:

2nd Reviewer -- Comment 1:

I have few remarks about the Introduction:

-"Weyl points are connected in momentum space by an unclosed Fermi arcs of topological surface states". It is ambiguous as the Fermi arcs are themselves the topological surface states. I would simply state "Weyl points are connected in momentum space by open Fermi arcs". Maybe, it is also worth to explain once that the Fermi arcs in the context of classical waves (bosonic and not fermionic) are not related to a Fermi surface but just to an equienergy line.

Response from Authors:

Following the suggestions of both Reviewer #2 and Reviewer #1, we have changed "Fermi arcs" to "surface-state arcs" in the revised manuscript.

2nd Reviewer -- Comment 2:

-The advantage of classical platform over real materials is argued to be in the absence of interactions. I do not agree as nonlinearities are the counterpart of interactions for these systems. The authors could elaborate further on why nonlinearities can be neglected in their sample. Alternatively, the concrete advantage of these platforms could be highlighted: the absence of intrinsic spin-orbit coupling.

Response from Authors:

We thank the reviewer for this good point.

We agree with the reviewer that nonlinearities in the classical platform play the role of interactions. However, for airborne sound waves, the nonlinear phenomena can occur only with an extremely strong sound pressure (> 135 dB, as in *J. Appl. Phys.* 120, 145105 (2016)). In our experiment, the sound pressure is very weak (< 35 dB). Therefore, nonlinearities are negligible. This is also consistent with the previous experimental works on other acoustic topological phases (e.g., *Nature Physics* 12, 1124 (2016); *Nature Physics* 14, 30 (2017); *Nature* 560, 61 (2018)).

We also agree with the reviewer that one of the advantages of these platforms is the absence of intrinsic spin-orbit coupling.

We have made two revisions accordingly. First, we have rephrased the description in the introduction, on page 3, starting from line 61, which now reads as:

“In comparison to the condensed matter systems where the nodal surfaces usually are fragile to spin-orbit interaction effects³⁰ or coexist with the other trivial bands, the classical-wave systems are advantageous in realizing the 2D nodal surfaces, as the structure of each unit cell can be precisely tailored and the resulting bandstructure is easily probed at various frequencies, and also because classical waves experience no intrinsic spin-orbit coupling.”

Second, we have explained the negligible nonlinearity in our acoustic system in the Methods, as follows:

“Besides, due to the limited sound pressure in our experiment, the nonlinear effects are negligible, as manifested in previous works^{9,19,21}.”

2nd Reviewer -- Comment 3:

-The simultaneous presence of Weyl points is briefly mentioned in the introduction. Their necessity is justified only later via the Nielsen-Ninomiya theorem. Such explanation should be provided already in the introduction.

Response from Authors:

Following the reviewer's suggestion, we have added an explanation in the introduction, which reads as

“Interestingly, in the acoustic/photonic systems^{29,33}, there are two linear Weyl points paired with the topological nodal surface and the total topological charges of all bands are zero, being consistent with the Nielsen-Ninomiya theorem³⁵.”

2nd Reviewer -- Comment 4:

The sample's design is based on the tight-binding model proposed in Ref.[29]. There, it is argued that a strong direct hopping among different layers can cause a transition from a nodal surface to nodal points. In the manuscript only the radii of the channels mediating the couplings are stated. The small size of the direct channels suggests that the authors are in the right regime of the parameter space. Would it be possible to extract the hopping strength mediated by this channel

by mapping the low energy description of the acoustic crystal around the nodal surface to an effective tight binding model?

Response from Authors:

Actually, our work is beyond the tight-binding model (TBM). We start with the TBM proposed in Ref. [29], and optimize the structure by maintaining the symmetry group. The resulting design, as shown in Fig. 1, is different from the TBM design—the acoustic waves in our design are never “tightly bound” to any site. This situation is general in the design of photonic/phononic crystals, since the bandstructures of photonic/acoustic crystals are due to the multiple Bragg scattering, and, in most cases, they cannot be described by the TBM (the photonic design in Ref. [33] is also beyond TBM). Therefore, in photonic/acoustic crystals, symmetry analysis is usually adopted to explain the band topology (e.g., the first proposal of topological photonic crystal by Haldane et al. in PRL 100, 013904 (2008)).

Our symmetry-based design has two advantages that facilitate experiment. Firstly, our design has a very large frequency range where the topological surface states exist (relative bandwidth ~20%), while the TBM design in Ref. [29] is small (relative bandwidth <2%). Secondly, there are z-oriented tubes running through the sample. The acoustic probes then can be inserted into the sample through these tubes, allowing the direct characterization of bandstructure.

Several other sentences have been added on page 6, starting from line 121, which reads as “Besides, we should note that in comparison to Xiao et al’s design²⁹, our design has a very large frequency range where the topological surface states exist (relative bandwidth ~20%), while the design in Ref. [29] is small (relative bandwidth <2%). Secondly, there are z-oriented tubes running through the sample. The acoustic probes then can be inserted into the sample through these tubes, allowing the direct characterization of bandstructure.”

2nd Reviewer -- Comment 5:

I assume that in Fig.1f the color of the bands corresponds to the local strength of the Berry curvature field. There should be a colormap to understand the numerical values of this figure. I would be careful in writing a Chern number for a gapless band structure. In the absence of a gap and in 3D the Chern number of a band is not well defined, although it is clear what the authors mean. I think that the colormap with the Berry curvature field value is sufficient for the authors' claim on the topological charge of the Weyl point and the nodal surface."we have verified that the depicted surface is indeed twofold degenerate": I find this sentence not useful in the absence of further clarifications on how the degeneracy has been assessed.

Response from Authors:

We are sorry for the confusion of color in Fig. 1f. The color of the bands in Fig. 1f in fact corresponds to the frequency, not the Berry curvature field. We have added a color bar in each panel to indicate it.

To avoid potential confusion, we have further removed the Chern numbers from Fig. 1f, and have revised the caption of Fig. 1, which reads as

“Each Weyl point has a topological charge of -1 and the topological nodal surface has a topological charge of +2, revealed via our numerical calculations (see Supplementary Note 1).”

Regarding the Chern number, we numerically calculated it by tracking the evolution of Wannier centers on the surface enclosing the band degeneracy (See Fig. S1). This approach has been widely adopted in literature [*Nature* 527, 495–498 (2015); *PRL* 120, 016401 (2018); *Nature Materials* 17, 978-985 (2018); *Nature Physics* 15, 645–649 (2019)].

Finally, we have deleted the statement that “we have verified that the depicted surface is indeed twofold degenerate”.

2nd Reviewer -- Comment 6:

Concerning the bulk characterization of the sample: Why did the authors choose to measure the pressure field only along one plane? A full characterization of the sample would have been more convincing. The data can then still be projected on 2d momentum planes or plotted at selected momentum points. In particular, it would have been possible to confirm experimentally the presence of a nodal surface in the entire 2d BZ at $kz=\pi$. I find the lack of a complete bulk characterization of the crystal the weakest point of the manuscript. I like how for these plots the results of simulations are presented but not overlaid to the measured data.

Response from Authors:

Actually, it is a general method to characterize 3D bandstructures with 2D projected ones (e.g., *Science* 349, 622-624 (2015); *Nature Physics* 14, 30-34 (2017); *Nature* 560, 61-64 (2018); *Science* 359, 1013-1016 (2018)).

The reason we project the 3D bandstructure to the 2D $kx-kz$ plane is because it is on the $x-z$ surface that we will measure the surface states and plot the surface arcs. This 2D projected bulk bandstructure provides a direct comparison with the 2D surface-state bandstructure that contains surface arcs.

2nd Reviewer -- Comment 7:

On the other hand, I find the characterization of the surface states very convincing and exhaustive. I particularly appreciate the experimental study of surface disorder that simultaneously proves the robustness of the surface states and their chirality. Concerning this part of the manuscript, I have only few remarks:

-The exact location of points A1,A2,A3,A4,A5 and A6 in momentum space should be stated. The name suggests that they are at $kz=\pi$ and $kz=-\pi$. Their location in kx is not clear. Have the data been symmetrized?

Response from Authors:

We thank the reviewer for considering that “the characterization of the surface states very convincing and exhaustive”.

A_1 to A_6 represent the momentum points $(\pi/a, 0.5\pi/h)$, $(0, 0.5\pi/h)$, $(-\pi/a, 0.5\pi/h)$, $(-\pi/a, -0.5\pi/h)$, $(0, -0.5\pi/h)$, and $(\pi/a, -0.5\pi/h)$, respectively. We have added a description in the caption of Fig. 3, which reads as,

“Here, A_1 to A_6 represent the momentum points $(\pi/a, 0.5\pi/h)$, $(0, 0.5\pi/h)$, $(-\pi/a, 0.5\pi/h)$, $(-\pi/a, -0.5\pi/h)$, $(0, -0.5\pi/h)$, and $(\pi/a, -0.5\pi/h)$, respectively.”

The data have been symmetrized due to the time-reversal symmetry preserved in our system. We have added a note in the revision, on page 8, starting from line 153, which reads as

“Note that due to the time-reversal symmetry preserved in our system, we have employed the inversion symmetry operator to make the measured dispersion symmetric.”

2nd Reviewer -- Comment 8:

-Fig.3b: Is there a reason why the measured band seems to extend below 4kHz, while the simulated one ends above it? Is this due to the boundary termination of the sample?

Response from Authors:

We thank the reviewer for his/her carefulness.

In the experiment, the measured momentum space is discretized with a limited resolution due to the finite sample size. The slight inconsistency between the measured and simulated results in Fig. 3b is due to the experimental error caused by the limited resolution.

We have added a discussion in the revision, on page 7, starting from line 149, which reads as “Note that the slight difference between the measured (Fig. 3(b)) and simulated (Fig. 3(c)) results is due to the experimental error caused by a limited resolution in momentum space.”

2nd Reviewer -- Comment 9:

-I think the termination of the sample should be clarified. The sample is closed with a plate and neither zigzag nor bearded termination are realized. Is this termination used also COMSOL?

Response from Authors:

The termination is shown in Fig. R3. The same termination is used in the simulations. We have added Fig. R3 in the Supplemental Information as Fig. S3.

Figure R3. (a) Top view of the termination chosen. Each hexagon represents a unit cell as shown in Fig. 1(c). (b)-(c) Different views of the sample surface.

2nd Reviewer -- Comment 10:

-Fig.3e&g: As opposed to the bulk measurements, here I would find more helpful if the simulated frequency contour of the projected Weyl points and nodal surface were overlaid to the measured data.

Response from Authors:

Following the reviewer's suggestion, we have put the simulated surface dispersion overlaid to the measured data, as shown in Fig. R4. One can see that the simulated and measured surface dispersions are in good agreement.

Figure R4. (a)-(f) Measured in-plane energy densities at frequencies ranging from 6.0 kHz to 7.8 kHz. For every plot, the horizontal axis (k_x) runs over $[-\pi/a, \pi/a]$, and the vertical axis (k_z) runs over $[-\pi/h, \pi/h]$. The green lines are the simulated surface dispersion for comparison.

We have added Fig. R4 in our Supplementary Information as Fig. S4.

2nd Reviewer -- Comment 11:

-Why does the agreement between simulation and measured data seem to improve at lower frequencies?

Response from Authors:

Due to the limited accuracy of the fabrication and measurement, there are slight differences between the simulated and measured results. As the frequency decreases and the corresponding wavelength increases, the acoustic dispersion becomes more insensitive to the details. Therefore, the agreement between simulation and measured data seem better at lower frequencies.

We have added a discussion on this point in our main text, on page 8, starting from line 156, which reads as

“Besides, due to the limited accuracy of the fabrication and measurement, there are slight differences between the simulated and measured results. As the frequency decreases and the corresponding wavelength increases, the acoustic dispersion becomes more insensitive to the details. Therefore, the agreement between simulation and measured data look better at lower frequencies.”

2nd Reviewer -- Comment 12:

-The frequency step of the measurement is 200Hz. Many figures look similar. Resolving 200Hz would amount to a quality factor of 30/40. What is the estimated quality factor of the sample? Is it reasonable to expect any observable difference over 200Hz?

Response from Authors:

The supported acoustic modes in our sample are running waves rather than resonance modes. Therefore, the quality factor that is usually used to characterize resonance modes does not apply

to our case. In our experiment, the frequency resolution depends on the recorder, which is around 1 Hz.

Besides, following the reviewer's suggestion, we have shown more frequencies from 6.20 kHz to 6.56 kHz with a frequency resolution of 40 Hz. As you can see from Fig. R5(a)-(f), the difference over 200 Hz is not obvious.

Figure R5. (a)-(f) Measured in-plane energy densities at frequencies ranging from 6.20 kHz to 6.56 kHz with a frequency resolution of 40 Hz. For every plot, the horizontal axis (k_x) runs over $[-\pi/a, \pi/a]$, and the vertical axis (k_y) runs over $[-\pi/h, \pi/h]$.

We have added Fig. R5 in our Supplementary Information as Fig. S5.

2nd Reviewer -- Comment 13:

Finally, there is a typo in the Experiments section of Methods. "The sample is manufactured via fabricated with additive manufacturing technique" should be "The sample is fabricated via additive manufacturing technique".

Response from Authors:

We thank the reviewer for careful reading. We have fixed this typo.

Reviewers' Comments:

Reviewer #1:

None

Reviewer #3:

Remarks to the Author:

The authors have addressed all my previous comments in great details and satisfactory manner. Therefore, I recommend publication in Nature Communications.

Response Letter to Reviewers

We are grateful for the constructive comments on this manuscript (NCOMMS-19-17253A) from all the reviewers.

In the text below, each of the comments from each reviewer is quoted in italics and is followed by our response.

COMMENTS FROM 2nd REVIEWER:

The authors have addressed all my previous comments in great details and satisfactory manner. Therefore, I recommend publication in Nature Communications.

Response from Authors:

We thank the reviewer for recommending the publication.